# Multi-Dimensional Quantum-like Resources from Complex Synchronized Networks

**DOI:** 10.3390/e27090963

**Published:** 2025-09-16

**Authors:** Debadrita Saha, Gregory D. Scholes

**Affiliations:** Department of Chemistry, Princeton University, Princeton, NJ 08544, USA; sahade@princeton.edu

**Keywords:** quantum-like, qudits, *k*-regular graphs, quantum resource

## Abstract

Recent publications have introduced the concept of quantum-like (QL) bits, along with their associated QL states and QL gate operations, which emerge from the dynamics of complex, synchronized networks. The present work extends these ideas to multi-level QL resources, referred to as QL dits, as higher-dimensional analogs of QL bits. We employ systems of *k*-regular graphs to construct QL-dits for arbitrary dimensions, where the emergent eigenspectrum of their adjacency matrices defines the QL-state space. The tensor product structure of multi-QL dit systems is realized through the Cartesian product of graphs. Furthermore, we examine the potential computational advantages of employing *d*-nary QL systems over two-level QL bit systems, particularly in terms of classical resource efficiency. Overall, this study generalizes the paradigm of using synchronized network dynamics for QL information processing to include higher-dimensional QL resources.

## 1. Introduction

Quantum computers typically encode information in qubits, the fundamental quantum analog of classical bits. However, many physical implementations of quantum hardware naturally support multilevel quantum systems beyond the standard two-level qubit structure. Examples include superconducting circuits with higher-energy excitations [1,2,3], trapped ions with multiple accessible electronic states [4], photonic systems utilizing different spatial or frequency modes [5,6,7,8], nuclear magnetic resonance [9,10], spin donors [11], and molecular magnets [12]. Qudits, the higher-dimensional generalization of qubits, encode information in a *d*-dimensional Hilbert space, thereby reducing the computational resources needed to simulate the same Hilbert-space dimension by a factor of log_2_(*d*).

Qudit gates reduce the number of entangling operations needed to implement important quantum gates like the Toffoli [3,13,14,15] and other unitaries, leading to improved circuit depths. Furthermore, qudits exhibit enhanced resilience to errors, as their extended state space allows for more robust encoding schemes in quantum error correction [16] and improved security in quantum communication networks [17]. Despite these advantages, scalable implementations of qudit-based quantum architectures remain an open challenge, since the additional energy levels introduce more pathways for energy relaxation and dephasing, resulting in shorter coherence times [18]. Furthermore, manipulating qudits for quantum information processing requires precise control over multiple transitions between energy levels that require sophisticated calibration techniques.

As an alternative, recent work suggests that complex classical networks with nonlinear interactions can give rise to exotic emergent states that exhibit decoherence-free behavior and quantum-like properties. While the field of quantum-inspired computing using classical resources remains in its early stages, these quantum-like features raise the possibility that such systems could, in principle, be adapted for storing and processing information in a manner analogous to quantum computation.

Synchronization in complex classical networks gives rise to emergent states that are isolated from the rest of the system’s eigenspectrum. This prevents the emergent states from decohering into other states, a desired feature for the storage and processing of quantum information. Furthermore, they may exhibit non-classical quantum-like (QL) probability laws, including interference of amplitudes [19,20,21]. These quantum-like properties of the emergent states make them promising candidates for realizing quantum-inspired computing resources. The building blocks suited for QL states are classical networks, possibly very complex, that we represent using random *k*-regular graphs that have expander properties.

In previous work [20], it was demonstrated that the eigenspectrum of the adjacency matrix of two coupled expander graphs exhibits emergent states that are isomorphic to the computational space of a qubit (see Figure 1a). Ref. [20] defined the formalism for quantum-like (QL) bits within the framework of classically synchronizing networks and showed that the emergent states of connected expander subgraphs are isomorphic to the Hilbert space of a qubit. Building on these findings, Ref. [22] introduced a general framework for constructing multi-QL-bit resources. In particular, it showed that the Cartesian product (□) of $N_{QL}$ such graphs yields a state space with dimensionality matching that of a register of $N_{QL}$ qubits (see Figure 1b). Furthermore, Ref. [21] also showed how arbitrary gates can be implemented by manipulating emergent states in these systems. However, to achieve a real advantage in simulating quantum systems or systems with quantum-like features [23], a higher number of these states is required to encode several important degrees of freedom. The number of physical resources required increases exponentially. However, multi-level resources could somewhat alleviate this issue and encode the same dimensional Hilbert space with log_2_(*d*) physical components.

In the present work, we build on the foundational work of Refs. [20,21,22,23] to formalize the concept of higher-dimensional quantum-like resources that we refer to as QL-dits. We present a theoretical framework for the construction of QL-dit resources using a family of *d* random *k*-regular graphs and their emergent spectra (panel (c) of Figure 1). The vertices represent the individual entities of the classical network—that could be a range of things like oscillators in an electric circuit, people in a social network, neurons in a neural network, and so on. The edges of these graphs model the correlations between these classical degrees of freedom. From the eigenspectra of these coupled graphs, we identify how the interplay between intra-graph and inter-graph coupling determines the presence of well-separated emergent states. These emergent eigenvalues form a map to computational basis states, offering a potential advantage for qudit-based quantum computation. The main goal of this work is to establish a general protocol on how to construct synchronizing networks that allow for multiple levels of emergent states that have quantum-like properties. We show how the intergraph couplings can be manipulated to give rise to a well-separated set of *d* emergent states that can be used as a basis for quantum information processing.

The structure of this paper is as follows: Section 2 introduces the formal construction of QL-dits using a family of *d* random *k*-regular graphs connected to each other by specific coupling geometries. In Section 2.1, we provide analytical expressions for the emergent eigenvalues across different intergraph coupling geometries. Section 2.2 shows how the corresponding eigenvectors can be represented. Section 2.3 presents numerical results demonstrating how the eigenvalues evolve with increasing intergraph coupling strengths. Section 3 shows how the Cartesian product of QL-dits can be used to represent the tensor product basis states of *N* QL-dits. Finally, Section 4 discusses the implications of these findings for quantum information processing using complex classical networks of oscillators and outlines potential directions for future research. Concluding remarks are provided in Section 5.

## 2. Quantum-like Dits

We begin by summarizing the key result in Ref. [20] to support and generalize the formalism developed here for mapping the state space of graph-based systems to that of a QL dit. The adjacency matrix *A* of a random *k*-regular graph *G* with *n* nodes exhibits a single isolated eigenvalue at *k*, well separated from a continuum of bulk eigenvalues [24,25]. Such graphs provide an abstract representation of complex classical networks with expander properties—for example, a network of coupled oscillators. The corresponding eigenvector encodes information about the synchronization pattern that appears in such networks represented by the graph. For the construction of a single QL bit, two such random *k*-regular graphs ($k_{1}=k_{2}=k$), $G_{1}$ and $G_{2}$, are weakly coupled via $C_{12}$ with an intergraph coupling valency, *l* [20]. The resulting adjacency matrix is given by:(1)$$\begin{array}{c} R=\left[\begin{array}{cc} A_{1} & C_{12}\\ C_{12}^{T} & A_{2} \end{array}\right] \end{array}$$
where $A_{1}$ and $A_{2}$ are the adjacency matrices corresponding to each subgraph and *T* means matrix transpose. The spectrum of R exhibits two emergent eigenvalues at k+l and k−l (Figure 1a), while the corresponding eigenstates are the symmetric and anti-symmetric combinations of the synchronized modes of $G_{1}$ and $G_{2}$, respectively. The associated eigenstates can be expressed as symmetric and anti-symmetric linear combinations of the eigenstates of $G_{1}$ and $G_{2}$. These emergent states provide a two-dimensional state space isomorphic to the Hilbert space of a two-level quantum system, making them suitable for encoding quantum information.

Furthermore, Ref. [22] extends the construction to multiple QL bits by forming the Cartesian products of graphs, whose emergent eigenstates map directly onto the tensor product structure of the Hilbert space for conventional qubit systems. Building on this, Ref. [21] introduces QL analogs of standard quantum gates like the single qubit Hadamard and the Pauli matrices, as well as two bit operations controlled NOT gate in quantum computation. The authors further show how these gates correspond to transformations of the underlying graph structure and the synchronization dynamics of the associated network of oscillators. Together, they establish both the construction of QL states for arbitrary number of QL bits and unitary transformations required to manipulate them for QL information processing.

The state space of a QL bit is a mathematical construct that arises from the underlying graph structure, and here we lay down the procedure to extend this construction to a higher-dimensional state space. We consider a system of *d k*-regular random graphs $\left\{G_{i}\right\}$, i∈{1,2,…,d} each with *n* nodes and adjacency matrices given by $A_{i},i\in \left\{1,2,\ldots ,d\right\}$. These subgraphs are undirected and loop-free. We introduce pairwise interactions by randomly connecting each node in $G_{i}$ to *l* nodes in $G_{j}$ forming coupling matrices $C_{ij}$ for 1≤i,j≤d. The overall adjacency matrix of the coupled system is as follows:(2)$$\begin{array}{c} R=\left[\begin{array}{cccc} A_{1} & C_{12} & \ldots & C_{1d}\\ C_{12}^{T} & A_{2} & \ldots & C_{2d}\\ \vdots & \vdots & \ddots & \vdots \\ C_{1d}^{T} & C_{2d}^{T} & \ldots & A_{d} \end{array}\right] \end{array}$$For simplicity, we choose all subgraphs with the same internal valency *k* and all inter-graph coupling matrices with valency *l*. The coupling geometry plays an important role in determining the emergent spectrum of this coupled system of graphs. In the limiting case of no coupling between the subgraphs ($C_{ij}={𝟘}_{n}$ for all i,j), R will have a single emergent eigenvalue at *k* with degeneracy *d*. This is illustrated in Figure 2a. On the other hand, when each subgraph, $G_{i}$, is coupled to every other subgraph $G_{j}$ through $C_{ij}$ with a fixed valency *l*, the largest eigenvalue of R is given by $\lambda_{0}=k+\left(d-1\right)l$ and the other emergent states correspond to $\lambda_{1}=k-l$ with multiplicity d−1 (shown in Figure 2b). The corresponding eigenvectors describe collective modes where subgraphs oscillate either in phase (symmetric mode) or out of phase (orthogonal fluctuations), and the degeneracies reflect the underlying symmetry of the coupling geometry.

In more general configurations, the degeneracies in the spectrum are lifted as the coupling geometry becomes less symmetric. We explore this effect by focusing on the intermediate coupling regimes such as nearest-neighbor connections and analyze two representative geometries (i) periodic and (ii) open chains of the *d* subgraphs. In the following section, we examine how the structure of intergraph coupling $C_{ij}$ shapes the emergent eigenstates in these cases.

### 2.1. Effects of Intergraph Coupling Geometry on the Spectra of QL-Dits

The emergence of non-degenerate eigenstates is critical for establishing a one-to-one correspondence with the computational basis of a qudit Hilbert space. For QL-dits, this requires that the *d* emergent states exhibit distinct eigenvalues, so that peaks associated with QL states remain identifiable in the density of states. In this section, we discuss the constructions that give rise to an orthonormal set of emergent eigenstates.

*Nearest neighbor coupling*: Each subgraph, $G_{i}$, is exclusively connected to its nearest neighbor subgraphs $G_{j}$ by coupling matrices $C_{ij}$. The adjacency matrix of the coupled system takes the form:(3)$$\begin{array}{c} R=\left[\begin{array}{cccc} A_{1} & C_{12} & \ldots & C_{1d}\\ C_{12}^{T} & A_{2} & \ldots & {𝟘}_{n}\\ \vdots & \vdots & \ddots & \vdots \\ C_{1d}^{T} & {𝟘}_{n} & \ldots & A_{d} \end{array}\right] \end{array}$$For simplicity, each $C_{ij}$ is chosen to be *l*-regular, allowing for non-zero diagonal entries. This allows, for instance, node *n* in subgraph $G_{i}$ to be directly connected to node *n* in $G_{j}$. As a result, the block structure of R admits a Toeplitz-like symmetry, with consistent subgraph and intergraph valencies (*k* and *l*, respectively). R is constructed to have a symmetric structure.

*Periodic Chain:* In a chain of *d* subgraphs, each subgraph is coupled to its two nearest neighbors, and periodic boundary conditions close the chain by connecting $G_{1}$ and $G_{d}$ through $C_{1d}$ and $C_{1d}^{T}$. This introduces cyclic symmetry, and the adjacency matrix adopts a block-circulant structure. The emergent eigenvalues take the following form:(4)$$\begin{array}{c} \lambda_{i}=k+2l\cos\left(\frac{2\pi i}{d}\right),\;i\in \left\{0,1,\ldots ,d-1\right\} \end{array}$$For a system of three coupled subgraphs (d=3), the eigenvalues are $\lambda_{0}=k+2l$, $\lambda_{1},$$\lambda_{2}=k-l$. The system is invariant under cyclic permutations of the three subgraphs. The symmetry group is $C_{3}$ whose irreducible representations give one symmetric mode and two degenerate antisymmetric modes. Figure 2c left shows the QL-dit of dimension d=6 periodic chain and the corresponding density of states, illustrating the degeneracies arising from its cyclic symmetry.

*Open Chain:* When the periodic boundary is broken (i.e., $C_{1d}=C_{1d}^{T}={𝟘}_{n}$), the system becomes an open chain. The emergent eigenvalues become(5)$$\begin{array}{c} \lambda_{i}=k+2l\cos\left(\frac{\pi (i+1)}{d+1}\right),\;i\in \left\{0,1,\ldots ,d-1\right\} \end{array}$$For d=3, this gives three distinct eigenvalues: $\lambda_{0}=k+\sqrt{2}l$, $\lambda_{1}=k$, $\lambda_{2}=k-\sqrt{2}l$. The lack of cyclic symmetry lifts the degeneracy seen in the periodic chain, yielding *d* distinct emergent eigenvalues. Figure 2d right illustrates the case of d=6 and the corresponding density of states of the associated adjacency matrix.

Importantly, in both periodic and open chains, the emergent eigenspectrum is bounded within the interval [k−2l,k+2l], irrespective of the number of subgraphs *d*. This is characteristic of the nearest-neighbor interactions between the subgraphs and has been seen for excitonic systems as well [25]. As *d* increases, the density of emergent eigenstates within this spectral window also increases, which may eventually reduce resolvability. This is shown in Figure 2 by plotting the $d^{th}$ emergent eigenvalues for a range of *d*. This suggests the existence of an optimal $d_{\mathrm{opt}}$ beyond which adjacent emergent peaks become indistinguishable. In particular, the lowest emergent eigenvalue asymptotically approaches k−2l for a fixed *l*, a behavior further examined in the next section.

*Limiting Case of QL-bits:* For d=2 (QL-bit), the periodic and open chain configurations yield the same spectrum. This equivalence arises because each of the two subgraphs has only one neighbor, rendering the boundary conditions indistinguishable. Consequently, the appropriate expression for the periodic case for d=2 is(6)$$\begin{array}{c} \lambda_{i}=k+l\cos\left(\frac{2\pi i}{d}\right),\;i\in \left\{0,1\right\} \end{array}$$
yielding $\lambda_{0}=k+l$ and $\lambda_{1}=k-l$. These results are consistent with previously reported QL-bit behavior [20,21] and thus provide the limiting case for the more general *d*-dimensional QL-dit construction. While this work focuses on nearest-neighbor coupling geometries, the emergence of *d* distinct eigenstates is also possible in systems with longer-range interactions, such as next-nearest-neighbor or fully connected geometries. However, these cases introduce additional symmetry considerations and may result in degenerate eigenstates. Analytical expressions for the emergent spectrum in such systems differ markedly from those derived here and are not discussed here further. In the present work, we focus exclusively on the case of nearest-neighbor coupling. However, it is also possible to obtain *d* emergent states in systems with extended coupling ranges, including interactions between next-nearest-neighbor subgraphs and beyond. The analytical expressions for the emergent eigenvalues in such cases deviate significantly from those derived for nearest-neighbor interactions and have degeneracies in the emergent eigenspectrum.

### 2.2. Structure and Symmetry of Emergent Eigenvectors

The eigenvectors of the QL-dit adjacency matrix (Equation (Equation 3)) corresponding to the emergent eigenvalues are represented in the d×n-dimensional space of the full graph structure of the QL dit. We denote the emergent eigenvectors of the individual *k*-regular subgraphs to be $a_{i}$, for i∈{0,…,d−1}. These vectors are defined in the *n*-dimensional space associated with the number of vertices in each random *k*-regular subgraph. We extend each state $a_{i}$ to a global vector $|u_{i}\rangle \in C^{dn}$ by padding the rest (n(d−1)) with zeros. Formally, $|u_{i}\rangle$ is defined as$$|u_{i}\rangle ={\left({𝟘}_{n},\ldots ,{𝟘}_{n},{\left(a_{i}\right)}_{n},{𝟘}_{n},\ldots ,{𝟘}_{n}\right)}^{T}$$
where the vector $a_{i}$ occupies the $i^{\mathrm{th}}$ slice of length *n*. This allows the emergent eigenvectors to be interpreted as superpositions of these localized modes. The $\left\{|u_{i}\rangle \right\}$ serve as a natural basis to express the emergent eigenvectors, $\left\{|\alpha_{i}\rangle \right\}$, of R as$$|\alpha_{i}\rangle =\sum_{j=0}^{d-1}c_{ij}|u_{j}\rangle$$
$\alpha_{i}$ is the eigenvector corresponding to the $i^{th}$ emergent eigenvalue and the coefficients $c_{ij}$ encode the projections $\langle u_{j}|\alpha_{i}\rangle$ on the subgraph modes across the full system. The coefficients satisfy the normalization condition $\sum_{j}{\left|c_{ij}\right|}^{2}=1$. The eigenvectors, $\left\{\alpha_{i}\right\}$, represented in this basis form an orthonormal set, where $\langle \alpha_{i}|\alpha_{j}\rangle =\delta_{ij}$ and spans a *d*-dimensional subspace in which any arbitrary unitary transformations of the QL-dit space can be defined and implemented. For illustration, the left panel of Figure 3 shows the structure of these coefficients for the open chain configuration of a QL-dit with d=6. The projection of each emergent eigenvector onto the subgraph blocks is visualized using a spider plot, highlighting the distinct amplitudes of projection onto each $|u_{j}\rangle$ for j∈{0,1,…,d−1}.

We further quantify the spatial structure of the eigenvectors using the Inverse Participation Ratio (IPR), defined for an eigenstate $\alpha_{i}$ of the adjacency matrix with components $\alpha_{ij}$ as(7)$$\begin{array}{c} IPR\left(|\alpha_{i}\rangle \right)=\sum_{j}{\left|\alpha_{ij}\right|}^{4} \end{array}$$This provides a measure of localization of the eigenstates across the nodes of the underlying graph structure for the QL-dit. High values indicate eigenstates that are concentrated on a few nodes, whereas low values correspond to delocalized states spread over many nodes. The right panel of Figure 3 illustrates the IPR for all eigenvectors for a QL-dit with d=6. The six eigenstates corresponding to the isolated eigenvalues exhibit notably low IPR indicating that they are delocalized across the graph structure. While the figure highlights a particular case, this qualitative behavior persists for arbitrary values of *d*.

### 2.3. Spectral Changes with Varying Intergraph Coupling Valency l

We investigate how the eigenvalue distribution, ρ(λ), of the QL-dit adjacency matrix evolves as a function of the intergraph coupling valency *l*, while keeping the subgraph size *n* and intra-subgraph degree *k* fixed. The number of nodes per subgraph is fixed at *n* and each subgraph is *k*-regular. Figure 4 illustrates the trend in the eigenvalues as a function of *l* for the particular case of d=6, though the results can be generalized for arbitrary *d*. For l=0, the subgraphs are uncoupled, and the spectrum consists of a single eigenvalue λ=k with *d*-fold degeneracy. As *l* increases, this degeneracy is lifted, and the eigenvalue at *k* splits into *d* distinct emergent eigenvalues (Equation (Equation 5)). According to Equation (Equation 5), these eigenvalues exhibit a linear dependence on *l*: the largest ⌊d/2⌋ increase linearly, while the remaining ⌊d/2⌋ eigenvalues decrease until they asymptotically approach the Wigner semicircle bound at $\tilde{\lambda }=2\sqrt{k-1}$. The change in the eigenvalue separation distance as a function of the coupling valency *l* is illustrated in Figure 4. It is important to note that Equations (Equation 4) and (Equation 5) for the lowest ⌊d/2⌋ emergent eigenvalues hold until they reach the value $\tilde{\lambda }$ after which it asymptotically follows the radius of the Wigner semicircle. For odd values of *d*, the middle eigenvalue at *k* remains constant for all values of *l*. All eigenvalues fall within the range [k+2l,k−2l].

Based on the analysis in Ref. [21], the optimal coupling for maximizing spectral resolution occurs at $l=\frac{k-\tilde{\lambda }}{2}$, which for k= is $l=\frac{k-\tilde{\lambda }}{2}\approx 8$. At this point, the spacing between emergent and random states is maximal. This result is crucial for generating well-resolved and distinguishable spectral states with potential utility in quantum-like information processing. These trends mirror those observed in QL-bits, laying the groundwork for a general theory of *d*-ary QL resources from synchronized networks.

The maximum value that *k* can be is n−1, for which the optimum value of *l* can be maximized to be ≈$\frac{n-2\sqrt{n-2}}{2}$. For these maximum values of *k* and *l*, the range [k+2l,k−2l] can be maximized for an optimal spacing between the *d* emergent states.

## 3. Cartesian Product of Graphs for Higher-Dimensional QL-Dit Spaces

To scale quantum-like (QL) information processing beyond a single QL-dit, it is necessary to construct composite systems comprising multiple QL-dits. Ref. [22] lays out the formalism for representing the state space of an arbitrary number of QL-bits based on the Cartesian product of single QL-bit graphs. Here, we extend the formalism to QL-dits and generalize it for arbitrary dimensional resources. As defined in Ref. [22], if $G^{\left(q\right)}$ denotes the graph system corresponding to the $q^{th}$ QL-dit, the Cartesian product of $N_{QL}$ such QL-dit graphs is given by$$\begin{array}{c} G=\overset{N_{QL}}{\underset{q=1}{□}}G^{\left(q\right)}. \end{array}$$The corresponding adjacency matrix for this Cartesian product of graphs can be expressed in terms of the individual adjacency matrices, $R^{\left(q\right)}$ of the QL-dit graph $G^{\left(q\right)}$ as(8)$$\begin{array}{c} R=\sum_{q=1}^{N_{QL}}\left[\overset{q-1}{\underset{p=1}{⨂}}1_{2n}\right]⊗R^{\left(q\right)}⊗\left[\overset{N_{QL}}{\underset{p^{\prime }=q+1}{⨂}}1_{2n}\right]. \end{array}$$The eigenspectrum of this adjacency matrix will have $d^{N_{QL}}$ emergent eigenvalues and the corresponding emergent eigenstates of these product graphs are tensor products of the emergent states of the constituent QL-dits. This is in accordance with the theory for the QL-bits as shown in Ref. [22]. Furthermore, the eigenspectrum of this adjacency matrix can be seen to have all possible eigenvalue sums of the $N_{QL}$ QL-dits. The emergent eigenvalues computed as the sum of the eigenvalues of the individual QL dits are also marked. If $\left\{\lambda_{i}^{\left(q\right)}\right\}$ represents the eigenvalue spectrum of the $q^{th}$ QL-dit, then the spectrum of G will have all possible $\sum_{q=1}^{N_{QL}}\sum_{j}\lambda_{i}^{\left(q\right)}$. This is shown in Figure 5 for the case of two QL-trits. The nine emergent states of the adjacency matrix corresponding to the Cartesian product of two QL-trit graphs are uniquely resolved and well separated from the random states. However, as $N_{QL}$ increases, the largest $2^{N_{QL}}$ eigenvalues start to merge with the random states. Therefore, the number of QL states at our disposal depends on the subgraph parameters *l* and *k* and can be tuned to purpose for a higher number of QL-dits.

## 4. Applications

Here, we provide examples illustrating how multi-level QL resources may be employed in information processing. The first involves encoding information within quantum probabilistic models relevant to cognitive science and decision-making frameworks. The second explores the use of QL-dits to represent the quantum state space of a prototypical quantum system, enabling the study of its dynamics within our QL formalism. While we focus specifically on the case of qutrits, the theoretical framework is developed for arbitrary local dimensions. These examples are intended as prospective applications of the formalism introduced in this work.

### 4.1. Expanding Quantum Cognition and Decision-Making Models

Information in our environment is inherently uncertain, and a central challenge in the behavioral sciences is to understand how the mind processes this uncertainty to form beliefs, make judgments, and arrive at decisions. Probabilistic models offer a natural framework for quantifying uncertainty and guiding inference under such conditions. However, classical probability theories [26,27], rooted in Kolmogorov’s axioms, often fail to account for observed violations of the law of total probability and the breakdown of additive rules in situations involving ambiguity or context. Notable examples include the question order effect [28], conjunction fallacy (as in the Linda problem) [29], and paradoxes such as the prisoner’s dilemma [30], where participant behavior deviates from the predictions of classical probability theory or heuristic explanations. In many such cases, the interference of mental states and context-dependent reasoning resembles features of quantum theory such as the non-commutativity of operations, superposition of beliefs, and interference in probability amplitudes [31]. These challenges have motivated the development of quantum probability models for cognition, which allow for contextual evolution [19,32,33], dynamic updates of belief states, and interference among competing mental representations [34,35,36,37].

These quantum-inspired frameworks leverage the vector space representation of quantum mechanics for representing information and concepts. For instance, the various possible attributes assigned to Linda in the conjunction fallacy can be modeled as vectors in a high-dimensional state space. Since human cognition is rarely binary, psychological responses often reflect graded responses, as captured by tools like Likert scales and semantic differentials [38], and representing such nuanced states requires a larger state space than what binary (qubit-like) models allow. Higher-dimensional quantum-like models, such as QL-dits, offer the capacity to encode multi-valued decision outcomes and represent ambivalent or uncertain states—e.g., “neutral,” “unsure,” or combinations like “both.”

The QL-dit formalism and their Cartesian product structure can provide a systematic way for expanding the vector space for encoding parameters involved in decision-making and cognitive scenarios. Then, each basis vector denotes a whole set of characteristics we need to consider in our decision space, and an exponentially expanding space will be able to encode more complexities of the decision-making paradigm. The question order effect in psychology—where a respondent’s answer to one question influences their response to a subsequent one—has been effectively modeled using principles from quantum theory, particularly the non-commutativity of measurements [19,39]. Analogous to the Stern–Gerlach experiment in quantum mechanics, where the sequence of spin measurements along different axes alters the outcome due to non-commuting spin operators, question order effects arise because cognitive measurements (i.e., questions) can act as non-commuting projection operators on a respondent’s belief state. This formalism, grounded in quantum probability theory, allows one to represent belief updating and interference in decision-making processes without invoking irrationality. By extending this framework to higher dimensions, QL-dits may support more faithful modeling of multi-alternative decisions and continuous belief dynamics, potentially offering a new class of cognitive models grounded in structured, quantum-like vector spaces.

### 4.2. Circuit Based Hamiltonian Simulation Using QL-Dits

Simulating many-body quantum systems remains one of the most promising and computationally demanding applications of quantum computing [40,41]. A major source of computational difficulty stems from the exponential growth of the many-body Hilbert space, formed as a tensor product of individual subsystem spaces [42]. Classical approaches such as tensor networks [43] and neural network quantum states [44] attempt to circumvent this challenge using compressed, low-rank representations. However, as quantum hardware scales up during the Noisy Intermediate-Scale Quantum (NISQ) era, limitations in qubit number and coherence time continue to constrain practical implementations [45]. An alternative route is to explore classical frameworks that provide robust emergent states that mimic quantum states. In this context, QL-dits may provide a promising platform. Building on earlier work with QL-bits [21,23,46], which reproduced qubit-based logic using synchronized networks, QL-dits extend the idea to emulate multilevel systems and more complex dynamics. Their expanded algebraic structure allows for richer gate operations and may provide a new classical approach to simulating quantum systems beyond the capabilities of two-level systems.

In Ref. [21] QL-analogs of standard single-qubit and two-qubit gate operations like the Hadamard, Pauli, and CNOT have been introduced for QL-bits. These operations allow for the definition of arbitrary unitary operations including both Clifford and non-Clifford, thereby enabling universal gate operations on QL bit states. In the broader context of quantum simulation and computation, a set of unitary matrices $U_{k}\in U\left(d^{n}\right)$ is considered a universal quantum gate set if their compositions can approximate any target unitary operator *U* on the Hilbert space $H_{d}^{⊗n}$ to arbitrary accuracy, as measured by a suitable norm [47]. For qudits, this universality has been similarly established using generalized gate sets [48,49]. The standard Pauli matrices, along with the identity, are the generators of the su(2) Lie algebra, which underpins operations in spin-$\frac{1}{2}$ systems. In a similar fashion, gates for qudits can be derived from the generalizations of the Pauli matrices for su(N) algebra, like the Heisenberg–Weyl matrices [50,51] or the Gell–Mann matrices [52,53] for three-level systems (qutrits). The clock and shift gates are the qudit generalizations of the Pauli $\hat{Z}$ and $\hat{X}$ operators, respectively, and form the building blocks for constructing arbitrary single-qudit gates. Thus, the notion of universality can be extended to the state space of QL-dits, as analogs of qudit operations using methods that have been defined for QL bits in Ref. [21]. This will allow for the simulation of unitary evolution of Hamiltonian dynamics of multi-level systems using a sequence of gate operations in the form of a QL circuit since any general Hamiltonian can be represented in these bases in the following manner:(9)$$\begin{array}{c} \hat{H}=\hat{H_{0}}I+\sum_{i=1}^{N^{2}-1}H_{i}\hat{S_{i}} \end{array}$$
where $\hat{S_{i}}$ of matrix dimension N×N are also the generators of the su(N) Lie algebra.

A natural question that arises is whether quantum many-body states can be encoded in these emergent states and their dynamics can be simulated using such classical networks. More specifically, can QL-dits provide any practical advantage in implementations of such quantum-inspired information processing? As discussed earlier (and in Section V of Ref. [21]), the graphs represent an underlying physical network, with the adjacency matrix R encoding two-body correlations between degrees of freedom. The dynamics of such networks is governed by the Kuramoto model [54,55] and synchronization leads to a globally stable state that remains robust over long time scales. Ref. [23] demonstrates how the collective steady state of a classical network can encode features of quantum information, suggesting that such synchronization dynamics may serve as a computational resource. Building on that framework, the long-term goal is to develop multilevel QL architectures that can simulate quantum systems efficiently, potentially offering lower implementation costs and greater robustness to noise compared to current quantum hardware.

### 4.3. QL-Dits in Topological Oscillator Networks

Topological systems use global structural properties to produce modes that are insensitive against local perturbations or disorder. Recent work in topological mechanics [56] and photonics [57] has shown that oscillator networks, when arranged with special geometries or connectivity patterns, can support protected edge or defect modes that remain stable even in the presence of disorder. Moreover, recent studies of topological synchronization in nonlinear oscillator lattices further highlight this connection, showing that synchronization can acquire topological protection through edge-localized modes [58,59]. Therefore, a further potential application of the QL-dit framework lies in the study of networks of oscillators that can provide topological advantage. These are features of the spectrum that are determined not by local details, but by global topological invariants of the underlying network structure such as winding numbers in one dimension [60], or Chern numbers in two dimensions [61]. Prominent examples of such oscillator networks include the Su–Schrieffer–Heeger (SSH) chain of oscillators [60], where alternating strong and weak couplings give rise to a winding number that enforces a zero-frequency edge mode. Embedding such topological structures within the QL-dit formalism could provide a natural platform to explore how topological protection manifests in quantum-like resources.

The core idea is to engineer an oscillator network whose bulk bands carry a nontrivial invariant; for example, SSH-like winding or a Chern number, while embedding that topological lattice on a substrate that is highly connected, i.e., exhibits expander properties. The topology forces existence of localized or edge QL states while the redundancy in connectivity can make the spectral gap practically insensitive to fabrication disorder and random rewiring. Extending to topological networks of oscillators could thus provide added robustness to the quantum-like states offering a classical analog to how topological quantum computing [62,63] uses nonlocal, protected degrees of freedom. In quantum systems, logical qubits are encoded in braiding patterns of anyons [62] while in oscillator systems, information can be encoded in the presence, absence, or phase of topologically localized modes. Because these modes are localized and protected, the information may be more resilient to perturbations.

## 5. Conclusions

A QL-bit is constructed from a pair of interacting networks (subgraphs), whose emergent states exhibit quantum-like properties that can be used to encode information and implement gate operations within a classical framework. This work generalizes such quantum-like resources to arbitrary dimensions by introducing QL-dits as higher-level analogs of QL-bits for quantum information processing. The adjacency spectra of these graph systems reveal well-separated eigenstates that map naturally onto the computational basis states of multilevel quantum systems. By constructing the Cartesian products of a system of QL-dits, we demonstrate how a scalable, tensor product structure of the isolated eigenvectors can be achieved, mimicking the behavior of conventional qudit systems.

The present framework, while promising, also suggests directions for further developments. Constructing the Cartesian product of $N_{QL}$ QL-dits leads to an exponential increase in the number of edges, reflecting its tensor product structure as given in Equation (Equation 8). Moreover, physical realizations of synchronization in such classical networks based on the Kuramoto model have been limited to tens or hundreds of oscillators as demonstrated across multiple experimental platforms, including a system of LC oscillator circuits [64,65] and spin torque oscillators [66,67]. This scaling therefore imposes a natural limit on feasible network sizes for both numerical and physical implementations and addressing this remains a central challenge. A promising strategy for mitigating this challenge, demonstrated by Ref. [23], is to exploit the fact that steady-state synchronization is governed by lower-rank approximations of the full network. In particular, Ref. [23] uses eigenvalue decomposition methods to approximate the Cartesian product of QL-bits, and these techniques can be generalized to the case of QL-dits. These approaches suggest that systematically reduced graphs can preserve the most relevant synchronization properties of larger networks, providing a direction for scalable implementations. A key future direction will involve employing rank reduction techniques to compress graph sizes while preserving spectral accuracy.

The QL-dit framework, nevertheless, opens several avenues for future applications. One direction is to develop a full suite of QL gate operations in higher dimensions, leveraging the algebraic structure of generalized Pauli matrices. These would allow for the simulation of arbitrary unitary evolutions in multilevel systems and facilitate the design of classical QL circuits for Hamiltonian dynamics. Another avenue lies in applying QL-dits to model cognitive processes, especially those involving multi-alternative decisions, graded belief states, or context effects such as question order sensitivity. While these possibilities are intriguing, it is important to note that the framework is currently best viewed as a testbed for demonstrating proof of concept studies. Ref. [23] shows entanglement generation by simulating Bell states using moderately sized networks, but scaling to practical implementations of QL algorithms remains challenging. In a similar manner, connections to models of cognition and decision-making are exploratory, yet the QL-dit formalism offers a natural vector-space expansion that could represent complex cognitive states more faithfully than binary models.

## Figures and Tables

**Figure 1 entropy-27-00963-f001:**
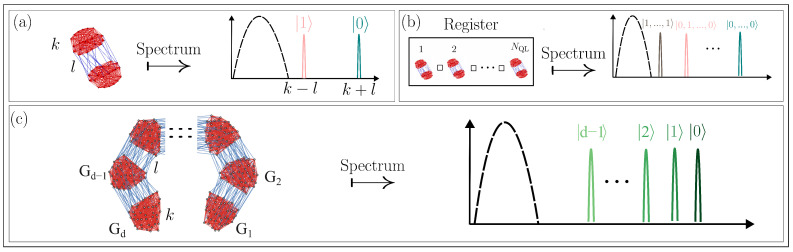
Schematic summary of previous work on QL-bit construction from coupled random *k*-regular graphs. (**a**) Two random *k*-regular graphs are weakly coupled via *l* intergraph edges, resulting in a density of states with two emergent eigenvalues isolated from the bulk of random states (shown as a dotted semicircle). (**b**) The emergent states of the Cartesian product of $N_{QL}$ such graph pairs correspond to the tensor product structure of multiple QL bits. (**c**) Conceptual illustration of the present work: generalization to multi-level QL-dits constructed from a system of *d* coupled subgraphs, forming a higher-dimensional emergent state space.

**Figure 2 entropy-27-00963-f002:**
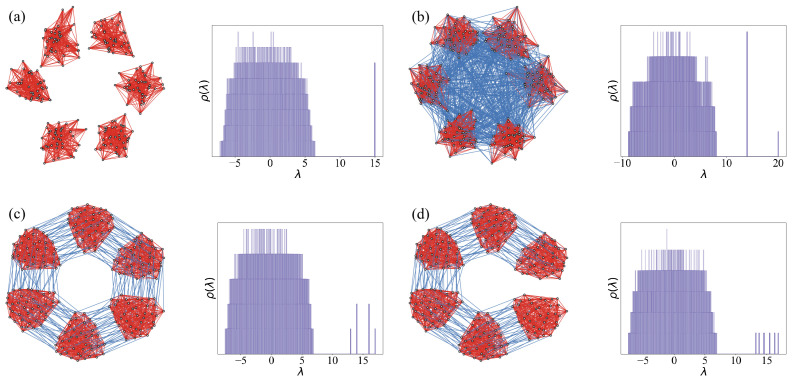
Graph structure and corresponding spectrum for configurations of the subgraphs with different inter-graph coupling geometries, *l*—(**a**) no coupling, (**b**) all to all coupling, (**c**) nearest neighbor coupling with periodic boundary conditions, and (**d**) nearest neighbor coupling with open boundary conditions between the subgraphs. In the case of periodic coupling, certain eigenvalues become degenerate due to the symmetric boundary conditions. In contrast, for the open chain configuration, the emergent eigenvalues are all non-degenerate. The density of states for both cases is shown. Each subgraph has n=40 vertices with an internal coupling valency k=15, and the intergraph coupling valency is set to l=1. The density of states is taken over 1000 independent realizations of the graph systems.

**Figure 3 entropy-27-00963-f003:**
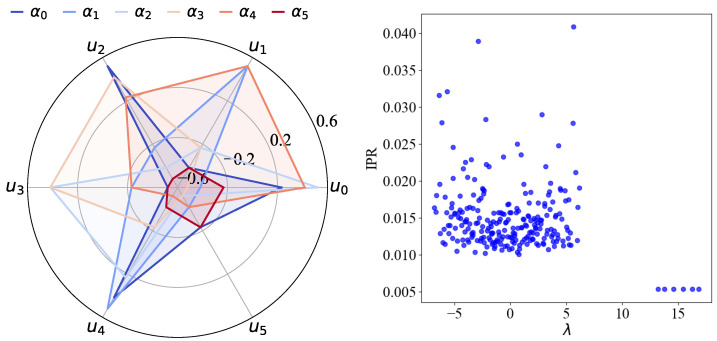
Emergent eigenvectors, $\left\{\alpha_{i}\right\}$, of the QL-dit system (d=6) projected onto the normalized emergent eigenvectors, $\left\{u_{j}\right\}$, of individual subgraphs (**left panel**). On the **right panel**, the IPR (Equation (Equation 7)) of the eigenvectors are shown as a function of the eigenvalues for the same QL-dit system. The lower IPR values associated with the isolated eigenvalues indicate that their eigenvectors are more delocalized compared to those corresponding to the bulk eigenvalues.

**Figure 4 entropy-27-00963-f004:**
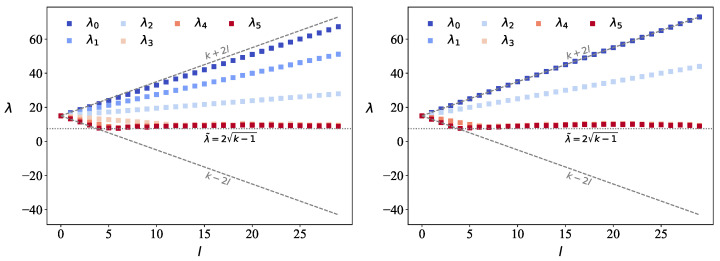
The dependence of ρ(λ) on *l* is shown for the case of d=6, with n=40, and k=15 for the open chain (**left**) and periodic chain (**right**). The larger $\left\lfloor \frac{d}{2}\right\rfloor$ eigenvalues can be seen to linearly increase with *l*, while the rest of the $\left\lfloor \frac{d}{2}\right\rfloor$ eigenvalues decrease. The latter ones also asymptotically go to $\tilde{\lambda }=2\sqrt{k-1}$.

**Figure 5 entropy-27-00963-f005:**
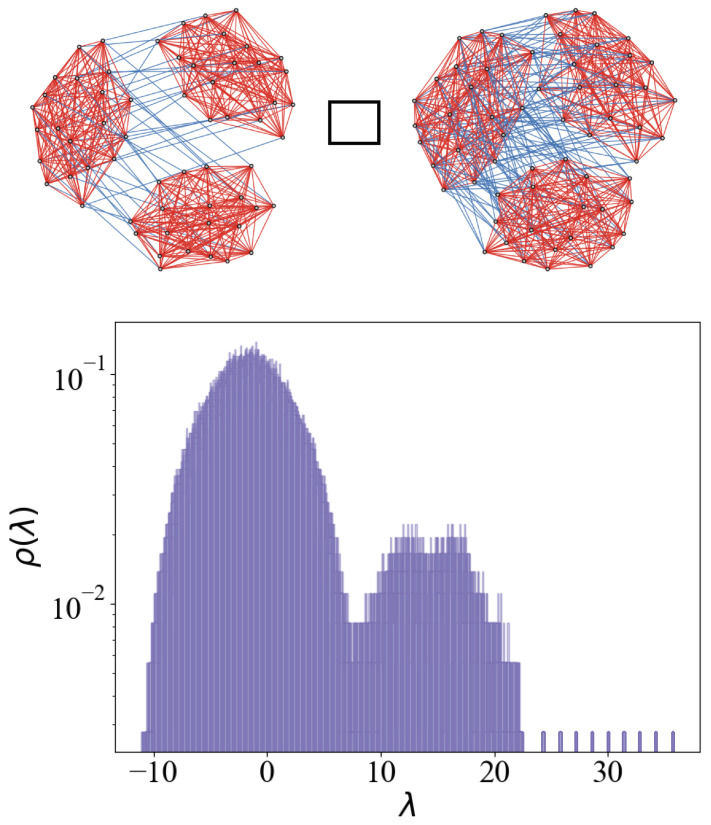
Density of states for the Cartesian product (**bottom**) of two QL-trits (**top**). The resulting $9=3^{2}$ emergent states are clearly resolved and non-degenerate due to the use of distinct intergraph coupling valencies *l* and $l^{\prime }$ for the two constituent QL-trits. The density of states is computed over N=100 independent graph realizations.

## Data Availability

The original contributions presented in this study are included in the article. Further inquiries can be directed to the corresponding author.

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
