# Peer review of "Multi-Dimensional Quantum-like Resources from Complex Synchronized Networks"

_entropy, 2025, doi:10.3390/e27090963_

Round 1

Reviewer 1 Report

Comments and Suggestions for Authors

In this work, the authors extend the recent works on quantum-like (QL) bits arising from the dynamics of complex synchronized networks. They extend the concept to higher-dimensional QL-dits, providing a generalized framework for multilevel QL information processing.

The authors construct QL-dits from k-regular graphs, define their state space via adjacency matrix eigenspectra, and achieve composite systems through Cartesian graph products, enabling a tensor-product-like structure analogous to qudits in quantum information theory. The work also discusses computational advantages of higher-dimensional QL resources and potential applications in both information processing and cognitive modeling.

The manuscript begins summarizing the previous works, especially Ref.[20] and Ref.[21], which I believe is very important to understand this relatively new “quantum-like” concept.   

To be honest, as a quantum information scientist frequently working on multilevel quantum particles, I only heard about this QL bits and I was not sure how well-structured concept it is. So, I am thankful for having the opportunity to review this manuscript.

Although one might think at first glance that extending a concept for two-level to multi-level systems is straightforward, this manuscript shows clearly that it is actually not, and why such an extension needs to be handled carefully and in detail.

Mathematical derivation is solid and easy to follow. Figures -especially the one on the emergent eigenvectors- are improving the presentation.

While the considered two applications in Section 4 are appropriate, I would suggest the authors to consider a third application, which is non-trivial topological systems consisting of oscillators, due to the significance of topological quantum information.  

Because 13% similarity looks a bit high, I checked the iThenticate report as well, and I confirm that there is no issue.

Finally, the manuscript fits well to the Section and the Special Issue.

Reviewer 2 Report

Comments and Suggestions for Authors

In their manuscript, “Multi-dimensional Quantum-like Resources from Complex Synchronized Networks,” the authors extend the emerging concept of quantum-like bits to general d-dimensional states, or quantum-like “dits.” They demonstrate how these states can be constructed by analyzing the eigenspectrum of coupled k-regular graphs and discuss potential applications of their framework.

I enjoyed reading this manuscript; it is well-written, and the central idea is clearly conveyed. However, one key aspect that appears to be missing is a characterization of the eigenvectors corresponding to the isolated eigenvalues. For now, only some preliminaries are discussed in Sec. 2.2. Even an illustration of the eigenvector structure on the underlying expander graphs would be a valuable and effective addition to the work.

I also suggest that the authors include a dedicated discussion of the framework’s potential limitations. For example, how challenging is it to physically implement the required framework – especially the Cartesian product of graphs? A note of caution regarding the use of these dits for NISQ applications, and particularly for models of “quantum” cognition and decision-making, would also provide essential context and balance to the manuscript.

A minor remark: the formatting of labels (such as Eq., Ref., Fig.) is inconsistent throughout the paper.

I would be happy to review a revised version that addresses these points and would reconsider the manuscript for publication at that time.

Round 2

Reviewer 1 Report

Comments and Suggestions for Authors

I thank the authors for taking into account the recommendations and revising their manuscript which was already very good. I recommend publication of this work.

Reviewer 2 Report

Comments and Suggestions for Authors

I find the authors' response thorough and convincing. I believe the manuscript is now suitable for publication.